# The Evolution of the Interactive Relationship between Urbanization and Land-Use Transition: A Case Study of the Yangtze River Delta

**Bo Niu** [1] , **Dazhuan Ge** [1,*] , **Rui Yan** [2] , **Yingyi Ma** [3] , **Dongqi Sun** [2] , **Mengqiu Lu** [4] and **Yuqi Lu** [1]

1. School of Geography, Nanjing Normal University, Nanjing 210023, China; 201302052@nnu.edu.cn (B.N.); luyuqi@njnu.edu.cn (Y.L.)
2. Institute of Geographic Sciences and Natural Resources Research, Chinese Academy of Sciences, Beijing 100101, China; yanrui20@mails.ucas.ac.cn (R.Y.); sundq@igsnrr.ac.cn (D.S.)
3. School of Architectural Engineering, Jinling Institute of Technology, Nanjing 211169, China; mayingyi@jit.edu.cn
4. School of International Economics and Trade, Nanjing University of Finance and Economics, Nanjing 210023, China; mqlu3201@nufe.edu.cn
* Correspondence: gedz@njnu.edu.cn; Tel.: +86-25-8589-1347

**Abstract:** In recent years, the impact of land-use systems on global climate change has become increasingly significant, and land-use change has become a hot issue of concern to academics, both within China and abroad. Urbanization, as an important socioeconomic factor, plays a vital role in promoting land-use transition, which also shows a significant spatial dependence on urbanization. This paper constructs a theoretical framework for the interaction relationship between urbanization and land-use transition, taking the Yangtze River Delta as an example, and measures the level of urbanization from the perspective of population urbanization, economic urbanization and social urbanization, while also evaluating the level of land-use morphologies from the perspective of dominant and recessive morphologies of land-use. We construct a PVAR model and coupled coordination model based on the calculated indexes for empirical analysis. The results show that the relationship between urbanization and land-use transition is not a simple linear relationship, but tends to be complex with the process of urbanization, and reasonable urbanization and land-use morphologies will promote further benign coupling in the system. By analyzing the interaction relationship between urbanization and land-use transition, this study enriches the study of land-use change and provides new pathways for thinking about how to promote high-quality urbanization.

**Keywords:** urbanization; land-use transition; interactive relationship evolution; PVAR model; coupled coordination

## 1. Introduction

Land-use change is a significant form of interaction and connection between human activities and the natural environment [1]. With the development of the economy, the impact of the land-use system on global climate change and global environmental change has become increasingly significant. In recent years, land-use change has become a hot issue [2–6], both in China and abroad. Grainger described land-use morphology as the general morphology of the actual land cover over a certain period of time, and proposed the concept of land-use transition (the changes in the land-use morphology over a certain period of time) [7], inspired by the "forest transition" hypothesis put forward by Mather [8]. At the beginning of the 21st century, the concept of land-use transition was introduced into China by Hualou Long, and it has been explored as a new means of land-use/cover change (LUCC) [9]. At the beginning of the introduction of land-use transition in China, it mainly referred to the time-series changes in land-use morphology affected by social and economic development [10]. With the deepening of the related research, the connotations of land-use

morphology became enriched, and its meaning expanded from the structure of land-use types to encompass the dominant and recessive morphologies of land-use. The former refers to the structure of the composition of the main land-use types in a region during a specific period of time, and the latter refers to the land-use morphology, relying on land-use dominant morphology, such as properties, functions, input and output [11,12]. The connotations of land-use transition have also been further deepened. At present, under the theoretical framework of the dominant and recessive morphologies of land-use, studies on land-use dominant morphology transition mainly focus on the problems of the evolution of quantitative land-use structural change (changes in the proportion of different types of land-use) [13] and space–time morphological characteristics (characteristics of land-use morphologies in terms of time series and spatial differences) [14]. Studies of land-use recessive morphology transition mainly focus on transitions in land-use function (main uses of the land) [15–17], land-use efficiency (output efficiency per unit of the land) [18], and land-use intensity (input per unit of the land) [19].

Multiple studies have identified different natural environmental and socioeconomic factors that function as the driving forces of land-use transition, and have carried out analyses of the driving mechanisms [20–22]. Among these factors, urbanization, as an important force promoting social and economic development, has a profound impact on the change of land-use morphology. Additionally, land-use transition also shows a significant spatial dependence on urbanization [23]. China's rapid urbanization began in the 1990s [24]. Under the current trend of economic globalization, urbanization not only promotes the rapid development of the social economy, but also remodels the form and morphology of land-use. The change in urban land-use structure, and the orderliness and rationality of land-use, are closely related to the urbanization process [23]. A review of related research on urbanization and land-use transition in recent years shows that current studies mainly focus on the role of urbanization in certain dimensions of land-use, such as land-use structure [25,26], land-use efficiency [27] and land-use intensity [28,29]. The effects of land-use transition mainly include the socioeconomic effect [30–32] and the eco-environmental effect [33–35]. However, few studies start from a systems theory perspective to undertake a comprehensive analysis of the two-way interaction and the mechanism between the urbanization system and the land-use transition system. Most studies only focus on the one-way impact of urbanization on land-use transition [23], or the impact of land-use transition [30–35]. The results of the study of Hualou Long and Yi Qu show that there is mutual feedback between land-use transition and land management, but this study does not address the interrelationships between land-use transition and urbanization [36].

An in-depth assessment of urbanization will reveal that economic efficiency is no longer the only goal of urbanization; urbanization is beginning to shift to high-quality, sustainable development. In this context, the relationship between urbanization and land-use also changes. Analyzing the two-way role between urbanization and the land-use system from the perspective of systems theory is of great significance to further understand both urbanization and land-use transition. From the perspective of land-use morphology, this paper argues that there is a two-way interaction relationship between urbanization and land-use transition (urbanization shapes land-use morphologies, and land-use morphologies feed back into and influence the process of urbanization), and that the interaction relationship tends to become more complex as urbanization progresses. Based on this, we focus on the two-way role between the urbanization system and land-use transition. The difficulty lies in how to reasonably measure and simulate the land recessive morphology [37] and how to quantify the two-way interaction between the urbanization system and the dominant and recessive morphology land-use systems. In view of this, this study takes the Yangtze Triangle—which has the highest densities of economic activity, population, and cities in China [38]—as its research object, and constructs a comprehensive index system to quantitatively describe its urbanization and the dominant and recessive morphologies of its land-use. This study also establishes panel vector autoregression

models (PVAR) and a coupled coordination model to describe the urbanization system and land-use morphology system from 2000 to 2020. In order to further enrich the theory of land-use morphology transition and clarify the relationship between urbanization and land-use transition, this paper analyzes the interactive relationships and mechanisms among the subsystems of land-use morphology.

## 2. Theoretical Framework

### 2.1. The Effect of Urbanization on Land-Use Transition

Land-use transition refers to a change in land-use morphology over a certain period of time [10], usually corresponding to a stage of transformation of social and economic development [11]. The driving forces of land-use transition can be divided into endogenous natural driving factors and exogenous economic driving factors according to their sources, wherein exogenous economic driving factors are generally considered the main factors driving changes in land-use morphology. Among the economic factors driving land-use transition, urbanization plays a significant role [39]. The main forms of urbanization include the spatial agglomeration of the population in cities (population urbanization), the development of social and economic industries to a higher stage (economic urbanization), and changes in the lifestyle and quality of life of urban residents (social urbanization). The above urbanization process has brought about corresponding changes in the areas of land-use structures, land-use forms, etc. (as well as urbanization). The expansion in terms of area of urbanized construction land causes the spatial morphology of land-use to change significantly [40]. In addition, changes in the recessive morphology of land-use, such as ownership, use, mode of operation, and input–output, are also reflected in agricultural land acquisition and conversion, and the transfer of rural collective construction land [41]. Therefore, urbanization causes land-use type change by promoting the expansion of urban space. The spatial morphology of land-use tends to be complicated, and the value attributes of land are also closely related to it. In addition, land-use morphology is also closely related to the stage of urbanization development, which makes the land-use morphology different in different regions.

### 2.2. The Effect of Land-Use Transition on Urbanization

The response of urbanization to land-use transition is related to the environmental effects of land-use transition, and the evolution of the spatial structure and the function of land-use. With the acceleration of urbanization, dramatic land-use transition in a region will inevitably lead to regional economic, social, and environmental changes [30–35,42]. Urbanization is an integrated process that encompasses higher stages of development in economic, ecological and social dimensions. Land-use changes in spatial structure, efficiency, and other dominant and recessive morphological aspects will directly affect the efficiency and quality of the development of society and the economy, which will also have a significant impact on the promotion of urbanization. Taking the problem of Urban Villages (A general term for the villages in the city) in China's urbanization as an example, to a certain extent, this is the result of unsuccessful land-use transition. This also reveals that inefficient urban and rural land-use and fragmental land spatial morphology will hinder the transformation of urbanization into the stage of high-quality development. The effects of land-use transition on resources and the environment are an important driving factor in promoting the transition to urbanization. An unreasonable land-use transition process (in the context of rapid urbanization, the rapid expansion of land for urban construction has put the regional ecology under duress) will worsen the resource and environmental problems, and lead to the aggravation of the ecological and environmental crisis [33–35], which will hinder the transition of the urban development strategy and the construction of resource/environment-friendly urbanization.

*2.3. Interactive Relationship between Urbanization and Land-Use Transition*

The relationship between urbanization and land-use transition is not a simple one-way relationship, but an interactive process. In terms of land-use dominant morphology, rapid urbanization inevitably leads to the expansion of urban space, which is mainly reflected in the rapid increase in the land area under urban construction, which aggravates the trend of land fragmentation in the absence of rational planning and control. The efficient utilization and sustainable transition of land-use are key to ensuring the orderly progress of urbanization. However, an unreasonable land-use structure and function system will become a significant obstacle inhibiting the development of urbanization. The urbanization process is closely related to the structural system and quantitative characteristics of land-use dominant morphology. The effectiveness of spatial expansion control in urbanization directly determines the direction and trend of construction land expansion, and significantly changes the quantitative relationship and structural characteristics of land-use dominant morphology. In the process of rapid urbanization, the structural transformation of construction land and agricultural land ensnares strong institutional and financial support for urbanization, which is an important guarantee to ensure the rapid advancement of urbanization. The above analysis shows that the dominant morphological change in land-use is an intuitive reflection of the spatial projection of the urbanization process, and it also affects the quality and process of urbanization development to a certain extent.

In terms of land-use recessive morphology, changes in land-use efficiency and intensity, and in the level of land-use function, effectively reflect the quality of urbanization. Relevant studies point out that the overall efficiency of land-use shows a downward trend against the background of rapid urbanization [43], reflecting that in the extensive urbanization mode, the value of land is not fully activated, and the problems of land recessive morphology transition and rapid urbanization become increasingly prominent. Similarly, disordered land-use patterns and management modes, inefficient land function development status, and extensive land inputs and outputs all become obstacles to further urbanization. In contrast, rational land-use structures and efficient land development patterns constitute an orderly land-use system and are conducive to the formation of a benign interactive relationship with the urbanization system, which promotes and coordinates the transformation between the two systems.

This study analyzes the interactive relationship between urbanization and land-use transition from the perspective of systems theory and constructs the theoretical framework in Figure 1, drawing on the concepts of the dominant and recessive morphologies of land-use to construct the interactive relationship between urbanization and land-use transition. Urbanization has reshaped land-use morphology and has an important impact on the dominant and recessive morphologies of land-use, which is manifested in changes in the spatial morphology, quantitative structure, efficiency, and function of land-use. Land-use morphology responds to urbanization and produces land-use transition, which, in turn, affects urbanization and its sustainable development. Based on this, the interaction mode between the urbanization system and the land-use system is discussed, and its interactive characteristics will become an important basis for promoting the development of urbanization quality. The coordinated coupling of land-use morphology and urbanization status is conducive to promoting the construction of high-quality urbanization. In contrast, a coupled antagonistic state of the two systems is a significant obstacle inhibiting the benign development of urbanization. Unreasonable urbanization will strengthen the conflict in the pattern of regional land-use morphology and become an important driving force promoting transition in the dominant and recessive morphologies of land-use.

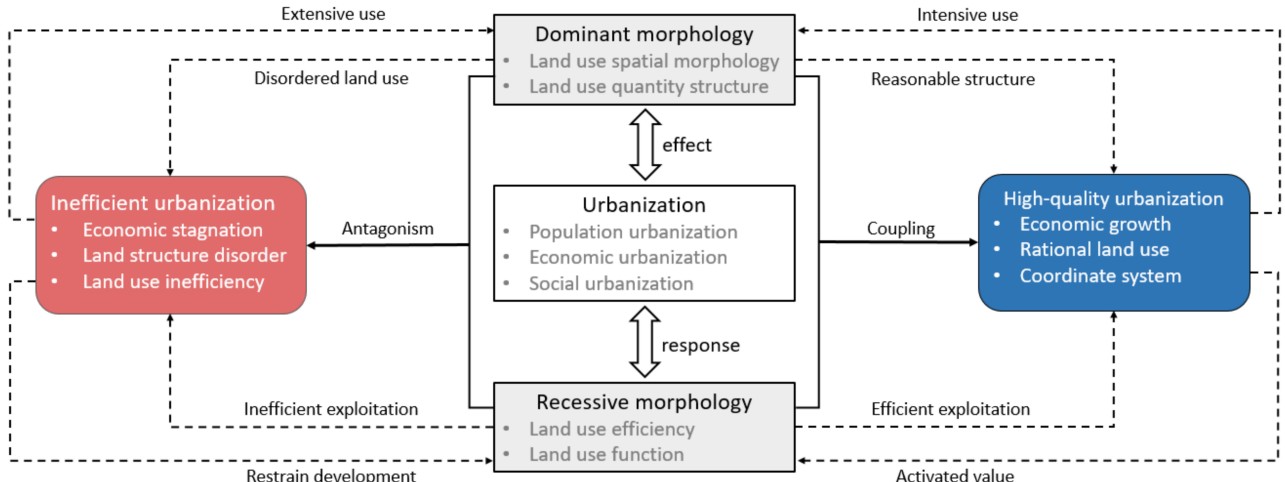

**Figure 1.** Analytical framework of interactive relationship between urbanization and land-use transition.

## 3. Data and Methods

### 3.1. Data Sources

To measure the level of urbanization, referring to the index system of related studies [44–46], this paper constructs a comprehensive evaluation system of urbanization and calculates the urbanization index (UI) of cities in the Yangtze River Delta region from the three dimensions of population urbanization (the spatial agglomeration level of the population in cities), economic urbanization (the development level of social and economic industries), and social urbanization (the quality of life of urban residents). The index data are from the China Urban Statistical Yearbook (2000–2020) and the local statistical yearbook (some of the indicators are not listed in the China Urban Statistical Yearbook, which are collated in the local statistical yearbooks of each prefecture-level city, such as the number of students in colleges and universities per 10,000 people, the number of beds in hospitals and health centers per 10,000 people, and the green coverage rate of built-up areas). It is worth noting that, as the relevant data from the 2020 Statistical Yearbook have not yet been released, the social and economic data from each local statistical yearbook in 2019 are applied to replace them. Due to the differences in the caliber of the statistical methods used to measure the urban population in some cities in earlier years, the urbanization rate of some cities increased nearly four times over five years. Therefore, this research applies the linear regression method to fit the urbanization rate of some cities in Anhui Province in 2000 to make the data stable.

As for the land-use data, in recent years, land-use remote sensing data have become an important support for LUCC-related studies [47]. This paper uses grid data for land-use (with a resolution of 1 km) in 2000, 2005, 2010, 2015, and 2020, which are available through the Resource and Environment Science and Data Center of the Chinese Academy of Sciences. On the basis of preprocessing the raster data, we calculate the relevant landscape pattern indexes using Fragstats to measure the land-use dominant morphology index (LUDMI). Combined with the raster data and socioeconomic data, we calculate the land-use recessive morphology index (LURMI).

### 3.2. Research Methods

Based on the main analysis methods and mathematical models of the article, we have drawn the analysis process into a flow plot (Figure 2).

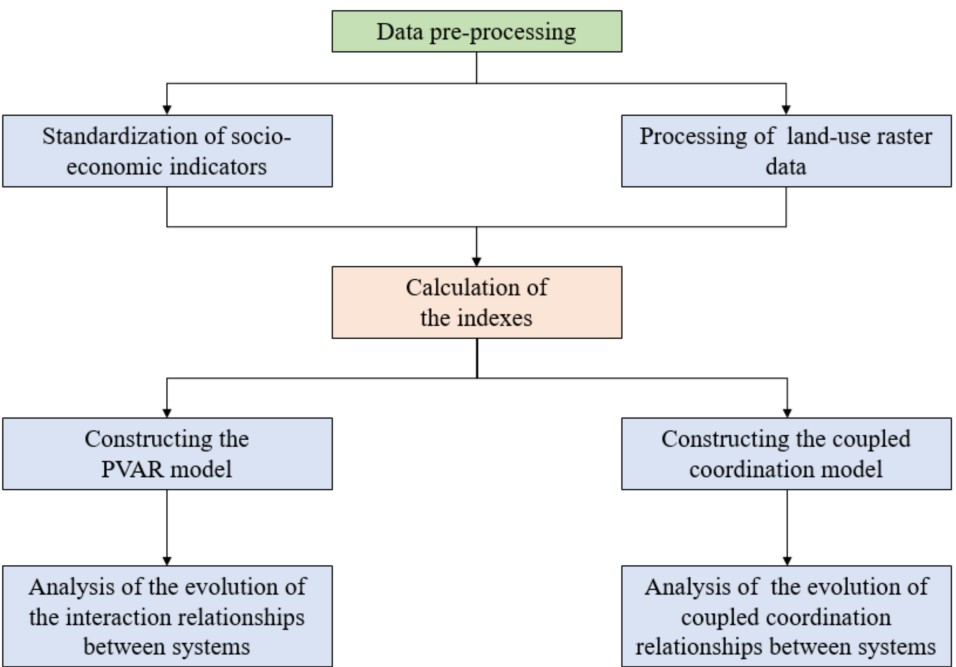

**Figure 2.** The process of data analysis.

### 3.2.1. Index System Construction and Weight Calculation

The purpose of this study is to explore the interactive relationship between urbanization and land-use transition, and the evolutionary mechanism that governs it. Based on the existing research [44], we select indicators from the three dimensions of population urbanization, economic urbanization and social urbanization to evaluate the comprehensive level of urbanization and then calculate the UI. From the perspective of land-use morphology, we study the dominant and recessive morphologies of land-use to describe the transformation of land-use quantitatively. Specifically, land-use dominant morphology (LUDM) is measured from the perspective of land spatial morphology (LSM) and land quantitative structure (LQS). By using the two aspects of land-use efficiency (LUE) and land-use function (LUF), this paper quantitatively simulates land-use recessive morphology (LURM).

Regarding the selection of the indicators, in the measurement of LUDM, we calculate the landscape fragmentation index using Fragstats to measure land-use spatial morphology from the perspective of landscape ecology. We also define the construction land structure index to evaluate the quantitative structure of urban land-use.

The calculation formula is $I_j = S_{con}/S_{non-con}$.

In the formula, $I_j$ is the construction land structure index of each city; $S_{con}$ is the area of urban construction land; $S_{non-con}$ is the area of urban non-construction land.

In terms of the measurement of LURM, we quantify LURM from dimensions of LUE and LUF. LUE is measured by calculating the proportion of the built-up area, investment in fixed assets per square kilometer of land, and the index of the comprehensive degree of land-use. Among these indexes, the comprehensive degree index of land-use characterizes the comprehensive degree of land development and utilization in the region by giving different types of land energy levels [48].

The calculation formula is $D_{ij} = 100 \times \sum_{r=1}^{n} A_{ij} \times C_r$.

In the formula, $D_{ij}$ is a comprehensive index of land-use degree in different years for each city; r means different land-use types; $A_{ij}$ is the proportion of different land-use types in the total land area; $C_r$ is the energy level of different land-use types. Furthermore, we quantify LUF using the aspects of the ecological, economic, and social functions of land-use. On the basis of the selection and calculation of the above indexes, the weight of each index is calculated using the entropy method, and then LUDMI and LURMI are calculated.

In terms of index assignment, the main methods can be divided into subjective and objective methods. To avoid the influence of subjective factors on the study, this paper applies the entropy method to determine the weight. The entropy value method refers to the definition of entropy in thermodynamics, which describes the degree of disorder of the system state with information entropy and calculates the index weight according to the value of information entropy. The main calculation steps are as follows [49]:

- Index standardization—Because the dimensions of different indicators are different, it is necessary to standardize the indicators. $x'_{\theta ij} = x_{\theta ij} / x_{max}$ is applied for the standardization of positive indicators, and $x'_{\theta ij} = x_{min} / x_{\theta ij}$ is used for the standardization of negative indicators. In the formula, $\theta$ represents the year, i represents the city, and j represents the indicator;
- Calculate the proportion of the index value—$Y_{\theta ij} = x'_{\theta ij} / \sum_{i=1}^{m} \sum_{\theta=1}^{n} x'_{\theta ij}$;
- Calculate the entropy of the index information $e_j = -k \sum_{i=1}^{m} \sum_{\theta=1}^{n} \left( Y_{\theta ij} \times \ln Y_{\theta ij} \right)$, in this formula, k is a constant term, and k =ln(mn);
- Calculate the redundancy of information entropy $d_j = 1 - e_j$;
- Calculate the weight of indicators—$a_j = d_j / \sum_{j=1}^{r} d_j$.

The weight of each measurement index of the urbanization system and land-use morphology is calculated by the above calculation formula, as shown in the Tables 1 and 2. After the index weight is obtained, the comprehensive measurement index of urbanization and land-use morphology is obtained by weighted summation.

**Table 1.** Comprehensive measurement index system for the level of urbanization.

| Dimension | Index | Index Weight |
|---|---|---|
| Population Urbanization | Urban population density | 0.093 |
| | Urbanization rate | 0.096 |
| | Per capita GDP | 0.090 |
| Economic Urbanization | Proportion of tertiary industry in GDP | 0.096 |
| | Total investment in fixed assets | 0.087 |
| | Regional passenger volume | 0.091 |
| | Total amount of social consumer goods per capita | 0.090 |
| | Average wage of employees | 0.092 |
| Social Urbanization | Education expenditure per capita | 0.082 |
| | Number of students in colleges and universities per 10,000 people | 0.088 |
| | Number of beds in hospitals and health centers per 10,000 people | 0.095 |

**Table 2.** Land-use measurement index system.

| LUMI | Dimension | Index | Index Weight |
|---|---|---|---|
| LUDMI | LSM | Landscape fragmentation index | 0.49 |
| | LQS | Construction land structure index | 0.51 |
| | | Proportion of built-up area | 0.16 |
| | LUE | Investment in fixed assets per square kilometer of land | 0.18 |
| LURMI | | Comprehensive land-use index | 0.16 |
| | | Green coverage rate of built-up area | 0.18 |
| | LUF | GDP per square kilometer of land | 0.15 |
| | | Population density | 0.17 |

### 3.2.2. PVAR Model

The traditional VAR model, proposed by Sims [50], is applied to predict and analyze the impact of random disturbances on variables in the time series, but the model does

not consider the problems brought by panel data. To address this issue, Holtz-Eakin [51] proposed the panel vector autoregression (PVAR) model. In recent years, following work to develop it by Love [52] and Lian [53], the model has matured and has been widely applied. The expression of the PVAR model is as follows:

$$Y_{it} = \sum_{j=1}^{m} \beta_j Y_{it-j} + \varphi_i + \omega_t + \varepsilon_{it} \tag{1}$$

In the formula, $Y_{it}$ is the column vector of UI and the land-use explicit and implicit form index; i and t represent the city and time, respectively; m is the order of delay; $\beta_j$ is the coefficient matrix of each lag term, which represents the degree of interpretation of $Y_{it}$; $\varphi_i$ is introduced to indicate the individual fixed effect, and reflects the heterogeneity of the cities in the Yangtze River Delta; $\omega_t$ represents the specific impact effect of each period and is the time effect vector; $\varepsilon_{it}$ is a random perturbation term.

### 3.2.3. Coupling Coordination Model

Coupling is originally a physical concept, which describes the phenomenon whereby two (or more) systems influence each other through interaction. The degree of coupling describes the degree of interaction between systems or elements, whereas the degree of coordination of coupling describes the trend of the system from disorder to order. The model's expression is as follows:

$$C = \sqrt{\frac{2 \times f(x) \times f(y)}{[f(x) \times f(y)]^2}} \tag{2}$$

$$T = \alpha f(x) \times \beta f(y) \tag{3}$$

$$D = \sqrt{C \times T} \tag{4}$$

In the formula above, C is the value of the degree of coupling, and f(x) and f(y) are the comprehensive evaluation scores of the two systems; T is the comprehensive evaluation index of the development of the two systems, $\alpha$ and $\beta$ are the undetermined coefficients; D is the coupling coordination index. To calculate the degree of coupling coordination, this paper divides it into three categories and six subcategories for evaluation and the classification results are summarized in Table 3.

**Table 3.** Classification of degree of coupling coordination.

| Category | D | | Subcategory |
|---|---|---|---|
| Coordinated type | 0.80–1.0 | T1 | High coordination |
| | 0.60–0.80 | T2 | Suboptimal coordination |
| Transitional type | 0.50–0.59 | T3 | Barely coordinated |
| | 0.40–0.50 | T4 | On the verge of disorder |
| Disordered type | 0.15–0.40 | T5 | Mild disorder |
| | 0–0.15 | T6 | Serious disorder |

## 4. Empirical Analysis of the Interactive Evolution

### 4.1. Two-Way Interaction Analysis Based on the PVAR Model

4.1.1. Index System Construction and Weight Calculation

In order to ensure the validity of the model results and avoid the pseudo-regression problem, and because the panel data applied in this paper only contain five periods, the HT test method, which is suitable for the unit root test of short panel data, is used to test the stationarity of the panel data.

The test results (Table 4) show that all variables except UI pass the HT test and that the data are stable. Based on the analytic ideas of this paper, two PVAR models need to be built to analyze the relationships between urbanization and land-use morphological systems (Model 1), and between urbanization systems and land-use morphological subsystems

(Model 2), respectively. Therefore, UI, LUDMI, and LURMI, as well as UI and the land-use morphological subsystem measures index, were tested covariantly using the Kao test. The test results (Table 5) show a long-term equilibrium relationship between UI and the remaining variables, i.e., the construction of the PVAR model can be conducted with the current data.

**Table 4.** HT unit root test.

| Variable | Stat. (Prob.), t |
|---|---|
| UI | 0.7642 (0.9997) |
| LUDMI | −0.4939 (0.0000) *** |
| LURMI | −0.6038 (0.0000) *** |
| LSMI | −0.8604 (0.0000) *** |
| LQSI | −0.7914 (0.0000) *** |
| LUEI | −0.6096 (0.0000) *** |
| LUFI | −0.3661 (0.0003) *** |

*** denotes the 1% statistical significance levels.

**Table 5.** Co-integration test based on the Kao test.

| | Model 1 Stat. (Prob.) | Model 2 Stat. (Prob.) |
|---|---|---|
| Modified Dickey–Fuller t | 4.8641 (0.0000) *** | 5.4363 (0.0000) *** |
| Dickey–Fuller t | 3.5435 (0.0002) *** | 5.1615 (0.0000) *** |
| Augmented Dickey–Fuller t | 1.6939 (0.0451) ** | 1.4211 (0.0776) * |
| Unadjusted modified Dickey–Fuller t | 1.3858 (0.0829) * | 1.3963 (0.0813) * |
| Unadjusted Dickey–Fuller t | −1.3953 (0.0815) * | −1.1374 (0.1277) |

*, **, and *** denote the 10%, 5%, and 1% statistical significance levels, respectively.

### 4.1.2. Model Order Determination and Granger Causality Test

The AIC test, BIC test, and HQIC test are also performed to determine the optimal lag order before establishing the PVAR model, and the test results are shown in Table 6. The results of the three tests show that the optimal lag order of Model 1 and Model 2 are all of order 1, so in this paper, order 1 is selected to construct the PVAR Model.

**Table 6.** Model lag order test.

| | AIC | BIC | HQIC |
|---|---|---|---|
| | −10.9054 * | −7.8874 * | −9.67947 * |
| Model 1 | −9.97569 | −5.8373 | −8.31419 |
| | −5.42103 | 0.848137 | −3.13814 |
| | −21.1123 * | −15.8537 * | −18.9762 * |
| Model 2 | −19.8522 | −12.3679 | −16.8474 |
| | 19.6186 | 31.3211 | 23.88 |

* denotes the optimal lag order calculated by each test method.

Granger causality tests are further performed for each variable in Model 1 versus Model 2 on the basis of determining model order (Table 7). The test results, shown below, show that only the Granger causality test of LSM for the remaining system variables does not pass the significance test, indicating that the fragmentation trend of land-use is not the Granger causality for the remaining variables, but that there is significant Granger causality between the remaining variables, which also further illustrates the rationality of the theoretical framework constructed in this paper.

**Table 7.** Granger causality test for variables.

| Equation | Excluded | Chi$^2$ | Df | Prob > chi$^2$ |
|---|---|---|---|---|
| | LUDMI | 70.778 | 1 | 0.225 |
| UI | LURMI | 1.4742 | 1 | 0.000 *** |
| | All | 71.282 | 2 | 0.000 *** |
| | UI | 4.7969 | 1 | 0.029 ** |
| LURMI | LUDMI | 14.925 | 1 | 0.000 *** |
| | All | 15.049 | 2 | 0.001 *** |
| | UI | 0.56121 | 1 | 0.454 |
| LUDMI | LURMI | 0.89521 | 1 | 0.344 |
| | All | 32.018 | 2 | 0.000 *** |
| | LUEI | 6.7271 | 1 | 0.009 *** |
| | LUFI | 4.0217 | 1 | 0.045 ** |
| UI | LSMI | 16.244 | 1 | 0.000 *** |
| | LQSI | 21.881 | 1 | 0.000 *** |
| | All | 29.718 | 4 | 0.000 *** |
| | UI | 5.6744 | 1 | 0.017 ** |
| | LUFI | 2.6833 | 1 | 0.101 |
| LUEI | LSMI | 8.7802 | 1 | 0.003 *** |
| | LQSI | 13.131 | 1 | 0.000 *** |
| | All | 17.048 | 4 | 0.002 *** |
| | UI | 6.2896 | 1 | 0.012 ** |
| | LUEI | 1.9892 | 1 | 0.158 |
| LUFI | LSMI | 6.196 | 1 | 0.013 ** |
| | LQSI | 6.5609 | 1 | 0.010 ** |
| | All | 8.7349 | 4 | 0.068 * |
| | UI | 0.0082 | 1 | 0.928 |
| | LUEI | 1.4912 | 1 | 0.222 |
| LSMI | LUFI | 1.2134 | 1 | 0.271 |
| | LQSI | 1.7681 | 1 | 0.184 |
| | All | 4.4417 | 4 | 0.350 |
| | UI | 4.2785 | 1 | 0.039 ** |
| | LUEI | 6.7495 | 1 | 0.009 *** |
| LQSI | LUFI | 4.2216 | 1 | 0.040 ** |
| | LSMI | 9.9698 | 1 | 0.002 *** |
| | All | 21.192 | 4 | 0.000 *** |

*, **, and *** denote the 10%, 5%, and 1% statistical significance levels, respectively.

### 4.1.3. Pulse Response Analysis

To further analyze the mechanism and path between the urbanization system and land-use transition, this paper conducted pulse response analysis using Stata 15.0 and took the trend of the pulse response function for further exploration.

Figure 3 presents the results of the pulse response analysis between the urbanization system and the land-use morphology system. Figure 3a,d shows that the elevation of LUDMI negatively inhibits the enhancement of the level of urbanization, that is, in both the LSM and land quantity structure, the enhancement of the land fragmentation level and increase in the structural ratio of construction land will hinder the enhancement of the comprehensive level of urbanization. The land-use pattern under rapid urbanization has resulted in an increase in the fragmentation of the landscape and the structural proportion of construction land, which will affect the urbanization system. According to the results (Table 8) of the cumulative pulse response, urbanization plays a positive role in LUDM, and the enhancement at the level of LUDM plays a negative role in urbanization.

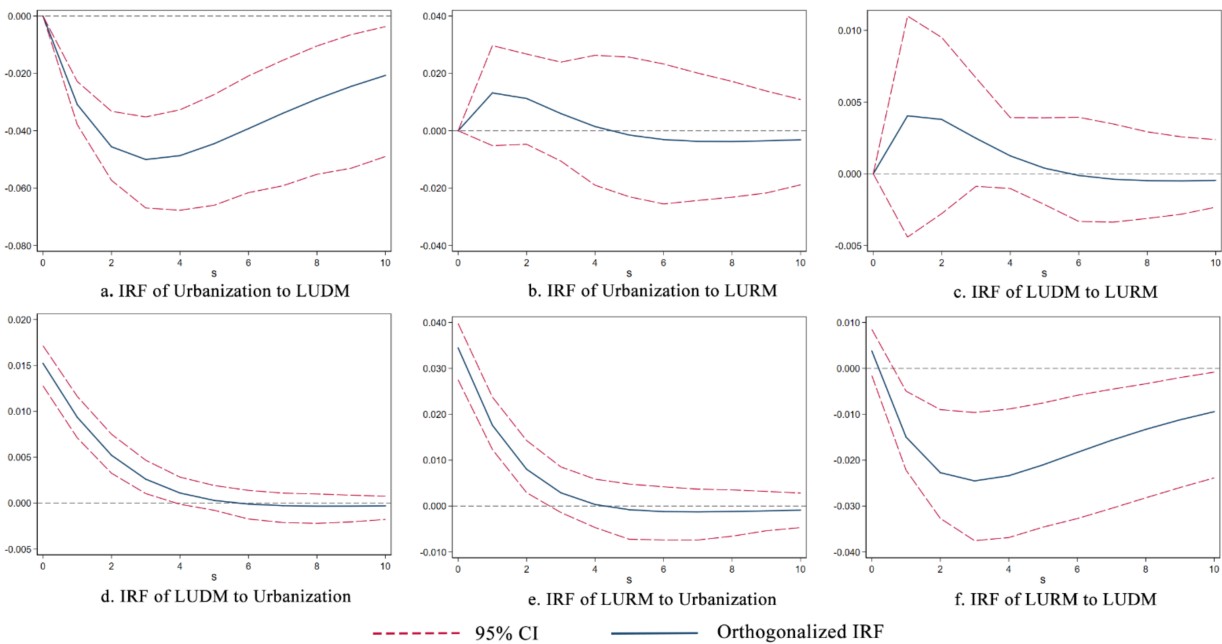

**Figure 3.** Pulse response results of Model 1.

**Table 8.** Cumulative impulse response values of Model 1.

| Response to | Urbanization | LUDM | LURM |
|---|---|---|---|
| Urbanization | 0.1079 | −0.3669 | 0.0133 |
| LUDM | 0.0325 | −0.0318 | 0.01 |
| LURM | 0.0569 | −0.1707 | 0.0613 |

Figure 3b,e show the relationship between urbanization and LURM. The results show that the relationship between urbanization and LURM manifests mutual promotion in the early stage, but that the positive promotion of one gradually weakens the other. However, after period 4, the interaction between the two turns into a weakly negative mutual inhibition relationship, and the impulse response of both becomes stable. Overall, the relationship between the urbanization system and the LURM system is positive and mutually beneficial.

According to the results of Figure 3c,f, the enhancement at the level of LURM, that is, the enhancement at the level of LUE and LUF, promotes landscape fragmentation and manifest a high structural ratio of construction land. However, after period 6, further increases at the level of LURM start to contribute to a trend toward the fragmentation of land and the continued growth of construction land, and LUDM shows a persistent negative effect on the enhancement of LURM. From the results of the cumulative pulse response, overall, the level of elevation of LURM promotes the level of elevation of LUDM, and LUDM transition, in turn, acts on LURM, inhibiting further improvements in it.

The above results show that the relationship between urbanization and land transition is not a simple linear relationship, but a complex non-linear relationship. When the system develops to a certain stage, the forces acting within the systems will change, e.g., the relationship between the urbanization system and LURM will change from one of mutual promotion to one of mutual inhibition.

Based on the results of Model 1, the relationships among urbanization, LUDM, and LURM are analyzed. In order to further discuss the specific mechanism of each system, Model 2 is established under the analysis framework of this paper.

According to the results of the impulse response analysis and the cumulative impulse response value of Model 2 (Figure 4 and Table 9), in general, in addition to the positive

effect of LUE on the development of urbanization, the cumulative effects of the fragmentation trend of LSM, the rapid expansion of construction land, and the improvement of LUF on urbanization are negative. Specifically, the enhancement of landscape fragmentation and the rapid expansion of construction land exerts a sustained negative effect on the enhancement of urbanization levels. LUE promotes the increase in urbanization level in the early period, but the positive effect gradually weakens and turns to a negative effect after period 6. There are fluctuations in the role of LUF in the level of urbanization, which, although initially shown as having an inhibitory effect on urbanization, continues to have a weak and stable positive promotional effect beyond period 4. In addition, urbanization systems have similar characteristics regarding the forces of each land-use morphological subsystem. Urbanization consistently promotes the fragmentation of regional land-use spatial morphology, and the forces gradually weaken, decaying to 0 after period 5. Moreover, the effects of urbanization on LQS, LUE, and LUF are all characterized by an early-period promotion and a late-period depression. Specifically, urbanization promotes the rapid expansion of constructed land-uses and the enhancement of LUE at the beginning, but restrains the development of both subsystems by the beginning of period 4; urbanization significantly promotes LUF in periods 0 to 2, but rapidly changes to have a continuously negative effect in period 3, suppressing the rising level of LUF.

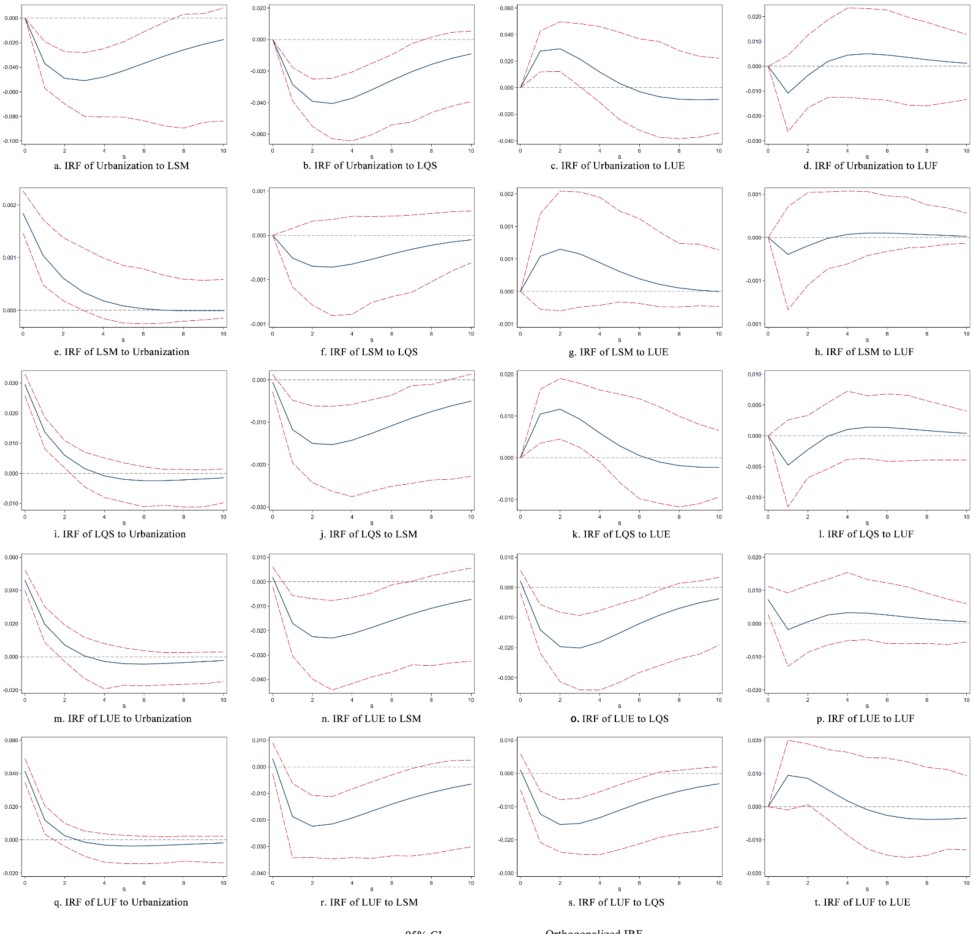

**Figure 4.** Pulse response results of Model 2.

**Table 9.** Cumulative impulse response values of Model 2.

| Response to | Urbanization | LSM | LQS | LUE | LUF |
|---|---|---|---|---|---|
| Urbanization | 0.0986 | −0.3602 | −0.2605 | 0.0577 | −0.0045 |
| LSM | 0.004 | 0.0054 | −0.0022 | 0.0028 | −0.0009 |
| LQS | 0.0376 | −0.1081 | −0.0413 | 0.0321 | −0.0089 |
| LUE | 0.0489 | −0.1574 | −0.1225 | 0.1018 | −0.0041 |
| LUF | 0.0337 | −0.1456 | −0.0945 | 0.0213 | 0.0537 |

Based on the exploration of the interactive relationship among urbanization and each land-use morphological subsystem described above, the relationship between the land-use morphological subsystems is analyzed by Model 2 in this paper (Figure 4). Inside the LUDM system, there appears to be an antagonism between the trend toward fragmentation of land and the expansion of construction land, suggesting that the expansion of construction land can, to some extent, slow the trend toward fragmentation in regional land in a landscape sense, and that the fragmentation of land is not conducive to the expansion of construction land. Inside the LURM system, where LUF is overall negative compared to LUE, the forces fluctuate and are not significant; LUE acts positively on LUF in the initial period, but changes to have a negative effect from period 5; this negative effect shows no sign of abating until stage 10. However, according to the results of the cumulative pulse response, the active force of LUE on LUF is positively promoted overall.

In terms of the effects of the LUDM subsystems on the LURM subsystems, the fragmentation of LSM and the expansion of construction land both negatively contribute to the enhancement of the levels of LUE and LUF; the effects peak in period 2 and gradually weaken, but persist thereafter. In terms of the effects of LURM on LUDM, increasing levels of LUF exert an overall weak inhibitory effect on land fragmentation versus the expansion of construction land, but there is some fluctuation in this effect; the enhancement of LUE makes the land morphology more fragmented, and the expansion of construction land is promoted in the early period and converted to a negative suppressing effect after period 6.

### 4.2. Analysis of the Coupling Coordination of Urbanization and Land-Use Transition

In the theoretical framework of this paper, we identify the direction, magnitude, and evolution of the relationships among urbanization, LUDM, and LURM by building PVAR models for pulse response analysis. In order to further analyze the evolution of the relationships among the three systems, this paper conducts a specific time and spatial analysis via a coupling coordination model.

#### 4.2.1. Time Series Analysis of Coupling Coordination

Based on the calculation of the degree of coupling coordination among urbanization, LUDM, and LURM for a more intuitive view of the area under study, we plot the degree of coupling coordination among the systems as a scatter plot (Figure 5), with the degree of coupling coordination between urbanization and LUDM and between urbanization and LURM as the X-axis and Y-axis, respectively, and those between LUDM and LURM as the Z-axis.

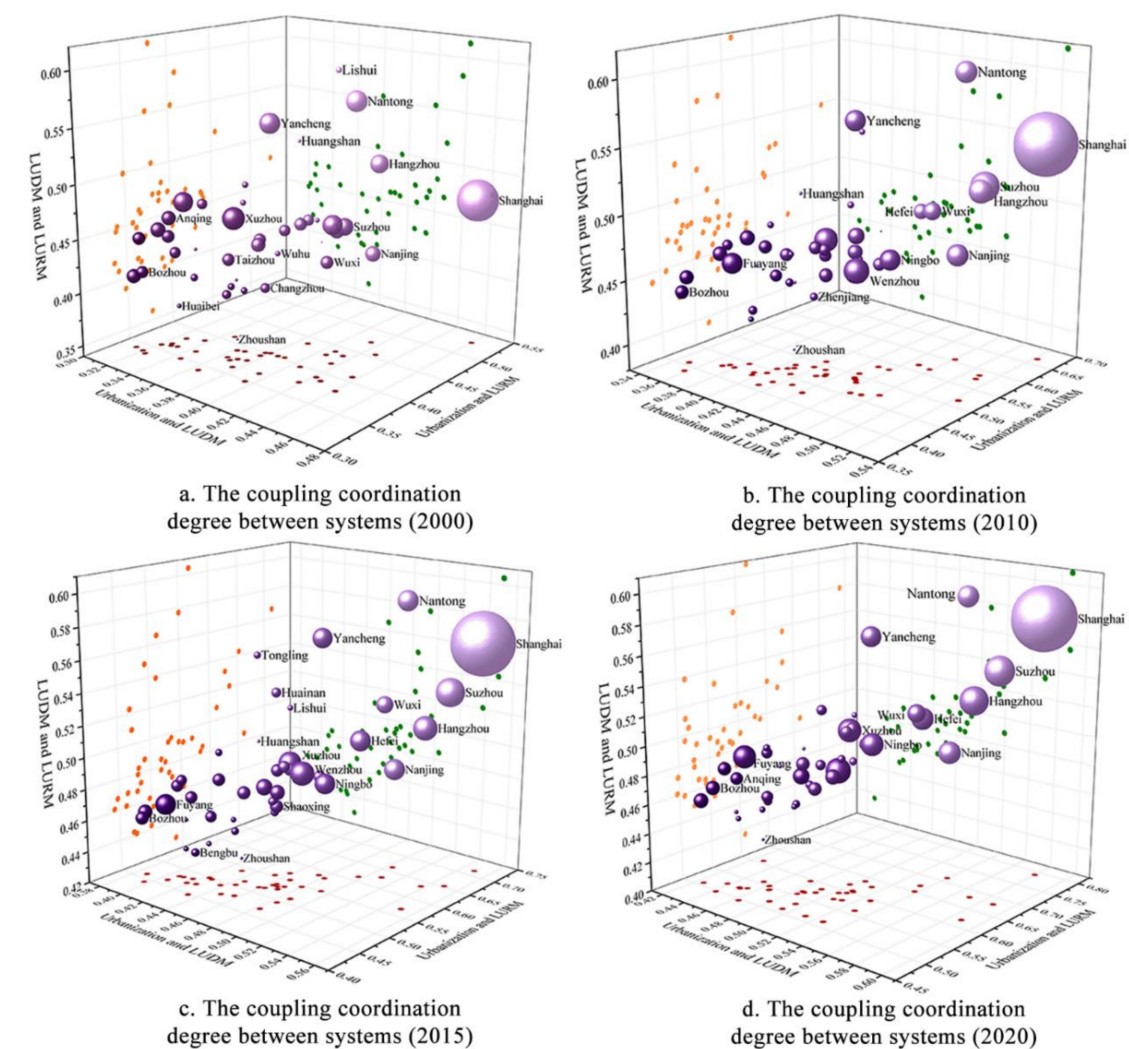

**Figure 5.** Time evolution of the coupling coordination relationship between urbanization and land-use morphology in the Yangtze River Delta.

In Figure 5, the size of the points represents the population size of each city, and the positioning of the point closer to the origin indicates a lower overall coupling coordination of the systems in that city, the relationship between systems is more antagonistic, and the development of system becomes moribund. The results show that the level of coordination between the urbanization system and the land-use morphology system varied among cities in the long triangle region in 2000, presenting discrete states in the plots. However, as time progressed, some regional hub cities showed large gains in both city size and system coordination, such as Shanghai, Suzhou, and Hangzhou. As can be seen from the projection of the scatter plots in the XY, XZ, and YZ planes, an overall increase in the level of coordination in the coupling between urbanization and land-use morphological systems in the long triangle occurred over the past 20 years, but the magnitude of the overall increase was not substantial, with only a few regional hub cities such as Shanghai and Suzhou having increased with respect to all aspects.

Furthermore, it is interesting to see from the graph that since 2000, there has been a convergence trend in the scatter plot, affected by the degree of coupling and coordination between the urbanization system and the land-use morphology system. This further shows that the urbanization mode and land-use mode play an important role in promoting and influencing the coordinated relationship between urbanization and land-use patterns, with results similar to the Matthew effect. A reasonable urbanization path selection and

land-use mode can allow some cities to not only improve their level of urbanization, but also to expand their scale; the development of each system also tends to be coupled and coordinated, realizing a benign interaction between the systems.

### 4.2.2. Spatial Analysis of the Coupling Coordination

To further discuss the spatial pattern of the coupling and coordination relationship between urbanization and land-use morphology, the computational results are visualized in this paper using ArcGIS, and Figure 6 shows the spatial evolution of coupling coordination between the urbanization and land-use morphology systems in the Yangtze River Delta from 2000 to 2020.

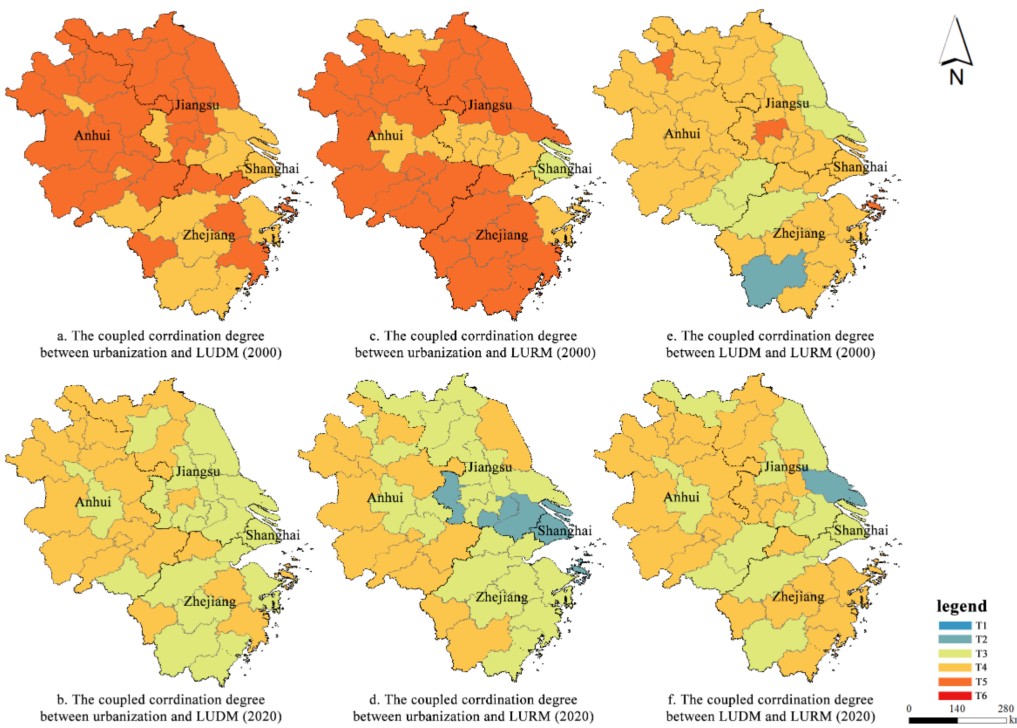

**Figure 6.** Spatial evolution of the coupling coordination relationship between urbanization and land-use morphology in the Yangtze River Delta.

In this paper, six types of degrees of coupling coordination are classified according to the calculation results. Figure 6a,b shows the spatial evolution pattern of the coupling coordination relationship between urbanization and LUDM in the Yangtze River Delta. In 2000, the coupling coordination relationships between urbanization and LUDM in over half of the cities in the Yangtze River Delta were mildly disordered (T5), while the rest were on the verge of disorder (T4). Specifically, cities belonging to T5 were mainly located in economically underdeveloped areas, such as Anhui Province and northern Jiangsu Province. Subsequently, the coordination relationship between the urbanization and LUDM of cities in the Yangtze River Delta began to moderate, and was advanced from coast to inland and from center to periphery.

Figure 6c,d shows the spatial evolution pattern of the coupling coordination relationship between urbanization and LURM. In 2000, almost all the cities' coordination relationships between urbanization and LURM in the Yangtze River Delta were in a state of antagonistic disorder, except Shanghai, which belonged to T3, and the cities in southern Jiangsu, eastern Anhui, and northeastern Zhejiang, which belonged to T4. In the subsequent development, the relationship between urbanization and LURM in the Yangtze River Delta area was gradually promoted from the east to the west. In the past 20 years, the level of coupling coordination between urbanization and LURM has been greatly improved in

Shanghai, southern Jiangsu, and eastern Zhejiang. In particular, the relationships of coupling coordination between urbanization and LURM in Shanghai, Nanjing, Suzhou, Wuxi, and Zhoushan were in a state of suboptimal coordination in 2020. The relationship between LUDM and LURM (Figure 6c,f) is not significantly related to the level of urbanization and economic development of cities. In the past 20 years, the coupling coordination relationship between LUDM and LURM in the Yangtze River Delta has not changed significantly. At present, the relationship between LUDM and LURM in more than half of the cities is on the verge of disorder.

## 5. Discussion

### 5.1. Interactive Evolution Analysis of Urbanization and Land-Use Transition

Based on the analysis of the relationship between urbanization and land-use transition in the Yangtze River Delta over the past 20 years by the PVAR model and coupling coordination model, we summarize the development process, as shown in Figure 7. This paper divides the evolution of the interactive relationship between urbanization and land-use transition into two main stages. S1 is the stage of early rapid urbanization, and S2 is the stage of accelerated rapid urbanization. On this basis, we put forward the stage of high-quality urbanization (S3), in which urbanization and land-use transition interact benignly.

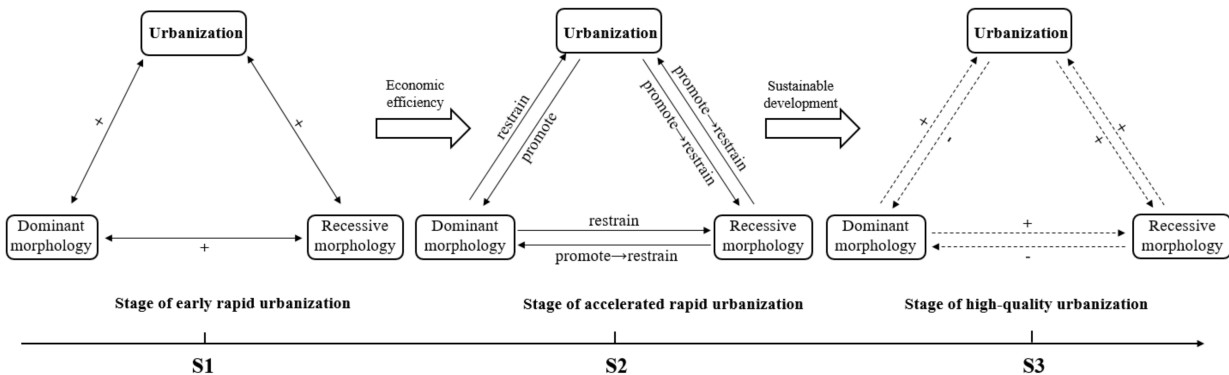

**Figure 7.** The bidirectional interactive response relationship between urbanization and land-use morphology.

In S1, the development level of each system is low, the relationship between the systems is simple, and the systems exhibit a synergistic promotional relationship with each other. Although urbanization promotes the rapid expansion of construction land and enhances the trend of land fragmentation, resulting in an increase in LUDMI, in the early stage, the extensive land-use model also brings some economic benefits, such that the level of urbanization increases. LUDM and LURM show a mutually promoting relationship. Meanwhile, a positive mutual promotion relationship is also observed between urbanization and LURM. However, it cannot be ignored that the coupling coordination relationship among the systems in this stage is mildly disordered or on the verge of disorder, which further illustrates that extensive urbanization brings about extensive land-use patterns that are unfavorable to the high-quality development of urbanization in the long run.

As rapid urbanization progresses, the relationship between urbanization and land-use transition moves into S2. In parallel with increasing urbanization, there is an urbanization orientation that solely targets short-term economic benefits, with the relationships among systems tending to be complex. For example, the trend toward land fragmentation caused by rapid urbanization is ultimately fed back to the urbanization system to inhibit further improvements in the urbanization level. The rapid expansion of construction land and the encroachment on other land do bring economic benefits and promote the improvement of the urbanization level in a certain period of time, but at a certain stage, this brings structural problems, which inhibit the development of urbanization in turn. In this stage, the interaction between urbanization and LURM has changed from positive to negative,

which also indicates that, in the long run, rapid urbanization will further affect the improvement of LUE and LUF, thus hindering the progress of spatial urbanization. Similarly, land fragmentation and the rapid expansion of construction land have a negative inhibiting effect on LURM, indicating that the unreasonable land-use structure is not conducive to the activation of land value. The effect of LURM on LUDM changes from promotion to inhibition, which further illustrates that the inefficient and low-functional land-use mode will aggravate the trend of urban construction land expansion and land fragmentation. In S2, the coordination relationship between the urbanization system and the land-use system improves, but this mainly reflects the improvement of the coordination relationship between urbanization and LURM. The levels of coordination relationships between urbanization and LUDM, as well as those between LUDM and LURM, do not improve.

This paper holds that urbanization and land-use transition are adapted to each other, and that there is a long-term balanced relationship between the two systems. A reasonable urbanization path and land-use pattern will form a benign interactive relationship, which will further deepen urbanization. In the long run, the extensive development mode brought about by the pursuit of short-term economic benefits will cause the system to fall out of order, show antagonistic effects among the systems, and hinder the efficient development of the social economy. Based on this, we have constructed an ideal stage of high-quality urbanization with sustainable development as the orientation. In S3, the patterns of urbanization and land-use have changed, and the structure of land-use is orderly and reasonable. Due to the intensive and efficient land-use under the high-quality urbanization mode, land fragmentation and the rapid expansion of construction land are restrained; urbanization effectively promotes the improvement of the LUF level and land output efficiency, which will also further affect urbanization and continue to promote its level.

Through the analysis of the evolution of the interactive relationship between urbanization and land-use transition, it is not difficult to see that the relationship between urbanization and land-use transition is not a simple linear relationship; the two impact and affect each other, and the relationship tends to become more complex with the development of the social economy to a higher stage. In order to realize the coordination relationship of systems in S3, urbanization path selection and land-use mode decision-making are the key issues. A reasonable urbanization path and an efficient, intensive land-use mode not only promote the sustainable development of urbanization to a higher stage, but also promote benign coupling in the relationship between systems.

### 5.2. Interactive Feedback between Rapid Urbanization and Land-Use Transition

Against the background of rapid urbanization, the extensive economic development and land-use pattern in the pursuit of short-term benefits are not only the result of this stage, but also factors influencing urbanization itself. Most cities in the Yangtze River Delta still have the problem that their land-use pattern is not suitable for higher stages of urbanization, which is not only an urban development problem in the Yangtze River Delta area, but also a common problem caused by rapid urbanization. In recent years, some cities in China have started to shrink for various reasons [54]. However, urban planning in China has long been established at the top level of the growth doctrine [55], and urban space still tends to grow more disordered. In addition, inefficient land-use, unreasonable land structures, and the occupation of agricultural land by urban construction land have further widened the urban–rural gap. The value of urban and rural land has not been fully activated. Further exploration of efficient utilization patterns is needed to tap the potential of land for regional coordinated development. These problems of urbanization and land-use are the real problems brought about by the rapid urbanization process. As the analytical logic of this article shows, these problems are both the results and the factors affecting the next stage of development. Therefore, how to achieve the benign development of high-quality urbanization between urbanization and land-use transition is a problem worthy of further discussion. As mentioned above, the process of rapid urbanization in China started in the 1990s [24], and with the aim of increasing the speed of urban

development, the rough development and land-use patterns have brought about urban problems such as Urban Villages and land-use fragmentation [56–59]. The results of this paper reveal the interaction between the urbanization system and the land-use system, and that land is not only a container and carrier of the city. From a systems theory perspective, the relationship between land and city is one of feedback and mutual influence. Taking a rational view of China's urbanization, as the understanding of urbanization deepens, the quality of urbanization becomes the first goal of development. How to achieve high-quality urbanization is the main issue. According to this study, we believe that land is not only a carrier of cities, but its participation in socio-economic development as a spatial element directly interacts with urbanization. Therefore, land-use transformation by means of land management and spatial remediation, and the formation of a positive interactive relationship with the urbanization system, are the keys to solving the problems brought about by China's rapid urbanization process, and achieving high-quality urbanization transformation.

In terms of the methodology, from a systems theory perspective, this paper takes the Yangtze River Delta as an example, creates two PVAR models and a coupled coordination model to analyze the evolution of the interactive relationship between urbanization and land-use transition, and holds that there is a long-term balanced relationship between the two whereby they influence and adapt to each other, which ultimately has a profound impact on the sustainable development of the social economy.

In terms of the results, this paper analyzes the evolutionary process of the interaction between land-use transformation and urbanization through the case of the Yangtze River Delta region, and analyzes the inter-system action mechanism. On the one hand, this study explores the interaction between urbanization and land-use transition from the perspective of the dominant and recessive morphologies of land-use, which enriches the connotation of land-use research. On the other hand, by dividing the interactive evolution stages of urbanization and land-use transition and analyzing the internal mechanism, we further clarify the importance of urbanization path selection and land-use mode decision-making, and provide a reference for urban development decision-making. In addition, the question still remains of how to achieve the coordinated transition of urbanization and land-use proposed in this paper, and ultimately achieve sustainable and high-quality urbanization— this issue has not been analyzed in detail in this paper, and is also worthy of further discussion.

## 6. Conclusions

This paper analyzes the interactive relationship between urbanization and land-use transition in the Yangtze River Delta from 2000 to 2020 from a systems theory perspective, and holds that there is a long-term equilibrium relationship between urbanization and land-use transition and that the two cause and affect each other, which ultimately has a profound impact on urbanization and the sustainable development of the social economy. The main conclusions of the empirical analysis of the Yangtze River Delta are as follows:

1. With the rapid development of urbanization in the Yangtze River Delta, the interaction between urbanization and land-use transition has changed from a simple positive interaction to negative inhibition between systems, and the interaction between systems has become more complex. Specifically, rapid urbanization intensifies the trend toward land fragmentation and promotes the rapid expansion of construction land, which hinders the further development of urbanization;

2. The structural problems brought about by rapid urbanization also make the interactive relationship between urbanization and LURM change. The relationship between them will inhibit both when it develops to a certain stage, which hinders the promotion of the overall level. This further reflects that the extensive development mode of rapid urbanization is not conducive to the improvement of land function level and LUE in the long run. Ultimately, urbanization itself will also be affected;

3.  Although the degree of coupling coordination between the urbanization system and the land-use system in the Yangtze River Delta region increased from 2000 to 2020, the overall level of improvement was not significant, and the system relationship of most cities was still on the verge of disorder. This indicates that, in the long run, the land-use transition problems brought about by the rapid urbanization mode will hinder the benign development of the system relationship;

4.  The coupling coordination relationship between urbanization and land-use transition in the Yangtze River Delta appears to be a convergence phenomenon, which also shows that a reasonable urbanization path and mode will promote benign coupling in the relationships between systems. This will ultimately make the city scale expand and the economy develop continuously; moreover, the systems will also achieve coordinated transition.

**Author Contributions:** Conceptualization, D.G.; methodology, B.N.; software, R.Y.; validation, B.N., Y.M. and M.L.; formal analysis, B.N. and D.G.; investigation, B.N. and R.Y.; resources, Y.L.; data curation, Y.M. and M.L.; writing—original draft preparation, B.N.; writing—review and editing, D.S.; visualization, B.N. and R.Y.; supervision, D.G. and D.S.; project administration, D.G. and M.L.; funding acquisition, D.G., Y.L. and M.L. All authors have read and agreed to the published version of the manuscript.

**Funding:** This research was funded by "the National Natural Science Foundation of China" (grant number 41901204, 42001125), "The Foundation of Humanity and Social Sciences of the Ministry of Education of China" (grant number 19YJCZH036, 20YJC790093), "China Postdoctoral Science Foundation" (grant number 2019M660109, 2021T140303), "Jiangsu Provincial Science Foundation" (grant number BK20190717), and "Jiangsu Provincial Social Science Foundation" (grant number 19GLC002).

**Institutional Review Board Statement:** Not applicable.

**Informed Consent Statement:** Not applicable.

**Data Availability Statement:** Not applicable.

**Conflicts of Interest:** The authors declare no conflict of interest.

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
