# Peer review of "The Evolution of the Interactive Relationship between Urbanization and Land-Use Transition: A Case Study of the Yangtze River Delta"

_land, doi:10.3390/land10080804_

Round 1
Reviewer 1 Report
The selected research topic is quite interesting and it has also been done in an appropriate way. However, some points are required to address in order to enhance the quality of the study.
Major
- Reliability of both data sets (Urbanization Index (UI), Land Use) is required to emphasize by providing clear sources. Just mentioning the name of the data source odes not scientific oriented. Please provide appropriate references or past research which uses the same data source.
- The value of the research output can enhance by discussing implications and applications.
Minor
- Figure 5: Names/types of the maps (a-f) must be included in the image caption.
- A manuscript line number is required to provide line-by-line minor comments. However, this manuscript is absented of it. Please read the author guidelines before the next submission.
Reviewer 2 Report
The authors analyzed the interaction between urbanization and land-use transition from the perspective of systems theory. The paper is informative and good structured. The title matches the content. The introduction and literature review provide sufficient background and include sufficient references. Research methods were described exactly.
There are some minor revisions are necessary before publication.
Abstract. Some journal policy requirements were not followed, namely: authors did not follow the style of structured abstracts.
Introduction: There is no clear research hypothesis in the introduction.
Reviewer 3 Report
Authors present an interesting manuscript regarding “The evolution of that interactive relationship between urbanization and land-use transition” in the Yangzte River Delta.
It is a very complex text that uses a high amount of data, analyzed in different ways, so it is difficult to understand. I recommend explaining in a more detailed way the processes carried out and also a lot of concepts that authors introduce.
Introduction
It includes a lot of concepts that should be explained with more detail to be understood. May be the authors can give an example of them and explain how they are related. We are not able to understand the other parts of the introduction if these concepts are not adequately explained
- land use morphology
- land-use transition
- land-use dominant morphology transition
- Recessive morphology transition.
- Land-use quantitative structural change
- Space-time morphological characteristics.
- Land-use function
- Land-use efficiency
- Land-use intensity
- Economic efficiency
Some of these concepts are explained afterward, but authors should explain them the first time they cite them.
Page 2: “However, few studies start from a systems theory perspective to undertake a comprehensive analysis of the two-way interaction and the mechanism between the urbanization system and the land-use transition system.” Present these studies and what are their main results
The importance of land use studies should be explained here, as well as other studies related with land use changes and urban growth in China.
2 Theoretical Framework.
As the different concepts are not well explained it is difficult to understand the different sections. For example, when authors said in page 3: “Land-use changes in spatial structure, efficiency, and other dominant and recessive morphological aspects will directly affect the efficiency and quality of the development of society and the economy,” or “Inefficient urban and rural land use and fragmental land spatial morphology will hinder the transformation of urbanization into the stage of high-quality development” can authors put an example of these assumptions, in which way affect them?
They also said “An unreasonable land-use transition process will worsen the resource and environmental problems and lead to the aggravation of the ecological and environmental crisis, which will hinder the urban development strategic transition and the construction of resource-environment friendly urbanization.” Can the authors explain what an unreasonable land-use transition process is? In which way would this assumption lead to an ecological aggravation or environmental crisis? In which way will hinder “the urban development strategic transitions”. Are there any studies to rely in these assumptions? And, Can the authors explain the meaning of “urban development strategic transition”.
Explain:
- what is and interactive process in terms of land use transitions.
- Unreasonable land-use structure (previously authors had mentioned “unreasonable land use transition process”)
When they said “The urbanization process is closely related to the structural system and quantitative characteristics of land-use dominant morphology”. Which are the quantitative characteristics and what the structural system is?
In page 4, can the authors explain the meaning of:
- Disordered land use patterns
- Inefficient land use function development
- Rational land-use structure
- Reasonable structure
- Efficient exploitation
3 Data and Methods
In the previous section, as the authors calculate an Urbanization Index regarding Population Urbanization, Economic Urbanization and Social Urbanization, the definition of these concepts have to be very clear. Authors have defined them in a slight way. An exhaustive explanation should be given. For example, in table 1 authors show the variables (index) used in each of them. The information of the variables should be given also in this section.
What information includes the index data from the China Urban Statistical Yearbook (2000–2020) and the local statistical yearbook?
I don’t understand this sentence: “Due to the differences in the caliber of the statistical methods used to measure the urban population in some cities in earlier years, the urbanization rate of some cities increased nearly four times over five years”. Can the authors give more information? What error does the information include? Have the authors made any analysis to verify or to correct these errors and to conclude that the best way to analyze (or correct) them is to carry out a linear regression?
3.2 Research methods
I recommend to explain the different methods in different paragraphs and to include the definition of the different concepts and the formulas.
Formulas should be explained. For example, when the authors said Scon/Snon-con this information should be defined.
“the weight of each index is calculated using the entropy method” Explain how.
It would be interest to include a figure with showing the flow of the analysis.
Results
In my opinion, analyses are so complex (too many and all of them too complex) that are difficult to understand.
Discussion
Need more work. Both methods and results should be compared with other studies (worldwide and in China).
Round 2
Reviewer 1 Report
Dear Authors
The manuscript has been improved
Reviewer 2 Report
The authors accepted all the comments in the revised paper version.
Reviewer 3 Report
The authors have improved the manuscript.